# How to design an art-science program? Self-reported benefits for artists and scientists in the VI4 artist-in-residence program

Skylar Cuevas[1], Qi (Kathy) Liu[1], Helen Qian[1], Max E. Joffe[2,3], Karisa Calvitti[4,5], Megan Schladt[5], Eric P. Skaar[4,5], Kendra H. Oliver[1,6,7,8]*

1 Communication of Science and Technology Program, College of Arts and Sciences, Vanderbilt University, Nashville, TN, United States of America, 2 Department of Psychiatry, University of Pittsburgh, Pittsburgh, PA, United States of America, 3 Vanderbilt Center for Addiction Research, Vanderbilt University, Nashville, TN, United States of America, 4 Vanderbilt Institute for Infection, Immunology, and Inflammation, Vanderbilt University Medical Center, Nashville, TN, United States of America, 5 Department of Pathology, Microbiology, and Immunology, Vanderbilt University Medical Center, Nashville, TN, United States of America, 6 Vanderbilt's Innovation Center the Wond'ry, Vanderbilt University, Nashville, TN, United States of America, 7 The Curb Center for Art, Enterprise and Public Policy, Vanderbilt University, Nashville, TN, United States of America, 8 Department of Pharmacology, Basic Sciences School of Medicine, Vanderbilt University, Nashville, TN, United States of America

* kendra.h.oliver@vanderbilt.edu

**Data Availability Statement:** We have uploaded the underlying data set to Qualitative Data Repository (https://data.qdr.syr.edu/dataset.xhtml?persistentId=doi:10.5064/F6SUSRIC).

## Abstract

While many new programs bridge the arts and sciences, a data-based examination of art-science program design can lead to more efficient programming. The Vanderbilt Institute for Infection, Immunology, and Inflammation Artist-in-Residence program is a virtual program that brings together undergraduate student "artists" and faculty-level "scientists" to generate science-art content. We have recruited over 80 artists and 50 scientists to collaborate in creating visual science communication content. Using self-reported data from both groups, we performed qualitative and quantitative analyses to define sources for negative and positive experiences for artists and scientists. We also identify areas for improvement and key features for in producing a positive experience. We found that artists participants had more positive responses about "learning something new" from the program than scientists. We also found that for both artists and scientists the length of the program and the virtual nature were identified as key features that could be improved. However, the most surprising aspect of our analysis suggests that for both "way of thinking" and "science communication to the public or general audience," were seen as significant beneficial gains for scientists compared to artists. We conclude this analysis with suggestions to enhance the benefits and outcomes of an art-science program and ways to minimize the difficulties, such as communication and collaboration, faced by participants and program designers.

## Introduction

There is a growing appreciation for science-based art programs and artist-in-residence experiences [1–3]. In 2019, the Vanderbilt Institute for Infection, Immunology, and Inflammation

**Funding:** This program is supported by the Vanderbilt Institute for Infection, Immunology, and Inflammation (VI4), Burroughs Wellcome Fund, The Wond'ry Center for Innovation, The Curb Center for Art, Enterprise & Public Policy, and The Communication of Science and Technology program within the College of Arts and Sciences. The author(s) received no specific funding for this work.

**Competing interests:** The authors have declared that no competing interests exist.

Artist-in-Residence (VI4 AiR) program was developed to produce visual science communication content. One of the main goals for the program was to create an opportunity for people to collaborate and gain exposure to new communities and visual communication approaches. We focused on recruiting undergraduate student "artists" and matching them with faculty-level "scientists" and their teams. Since that time, VI4 has brought together over 80 artists and 50 scientists resulting in over five accepted cover images, 15 publish paper graphics, and 118 images, animations, and other media for science conferences, websites, and social media. However, we were interested in identifying the benefits of this programming for our participants to improve the design.

It is unclear how to best design art-science programs and programmatically support art-science collaborations. Based on previous publications, partnerships appear to be most valuable when scientists and artists have a shared stake in the project [4,5]. It is also essential that programs facilitate the ability to jointly communicate, design, and critique the work [4,5]. Anecdotally, art-science experiences can prompt novel ways of thinking for artists and scientists and result in powerful visual science communication products. Although there is usually a focus on the public impact of the art, these experiences also transform the participants [6].

For scientists there is a lack of training in visual science communication, which has essentially been neglected within graduate-level curriculums. Estrada and Davis identified that visual materials have typically been treated as an add-on instead of being an integrated aspect of science communication [7]. Visual literacy, defined here as a holistic construct encompassing visual thinking, learning, and communication, is a critical ingredient in the effective communication of science among expert and lay audiences [8]. However, within peer-reviewed literature, written and oral presentation skills has dominated science communication research [9–13]. Less emphasis has been placed on enhancing the visual aspects of communication. One approach to correct this deficit is bringing together artist and scientist to collaborate on developing visual science communication content.

At present, support for the benefits of art-science partnerships is anecdotal [6,14]. There have been surprisingly few attempts to test the widely held assumption that engaging in the arts makes one more creative [15]. And while the visual arts have the potential to develop students' creativity, objectivity, perseverance, spatial reasoning, and observational acuity—all key science skills—it is not clear whether these skills are developed through art-science collaborations [15–17]. Past collaborations between artists and scientists suggest that combining these disciplines can have transformative effects for the individuals involved [18,19]. We believe that answering the lingering question of benefits for artists and scientists, respectively, in art-science collaborations can provide insightful design directions to accentuate professional growth for both artists and scientists.

Our program focused on producing all types of visual science communication content including images, graphics, and visual representations to communicate scientific knowledge. Current guidance on improving the visual aspects of science communication range from step-by-step-style instructions to hyper-focused aspects of data visualization [20]. The value of science communication in engaging the public is clear [21–24]; having tools and opportunities to increase the development and effectiveness of science visuals is imperative [25,26]. The practice for academic and research institutions to hire graphic experts is rare [27]. While there are some guides for improving various facets of visual communications, including graphic design principles, many of these publications, however, are limited in scope [28], or data visualization focused [20,27,29–31]. Art-science and AiR programs are one way to bring more discussion and attention to science visuals and visual resources.

The goal of this work is to identify the benefits and areas for improvement when designing art-science collaborative programs. We were interested in exploring this question by

comparing the artist versus scientist perspective in the VI4 AiR program. Within the methods we provide an overview of the program, participants, and analytic methods used. Using the 2019–2021 post-survey data from VI4 AiR artist and scientists our results reports the benefits for both groups, identifies how to better design the program, and discusses best practices in program design. We found that artists participants had more positive responses about "learning something new" from the program than scientists. We also found that for both the artists and the scientists the length of the program and the virtual nature were identified as key features that could be modified. However, the most surprising aspect of our analysis suggests that for both "way of thinking" and "science communication to the public or general audience," were seen as significant beneficial gains for scientists compared to artists.

## Methods

### Program development

**Recruitment of artists and scientists.** Table 1 shows institutions of artist and scientists that participated in the program over the last three years. We consider minority-serving status, NIH funding ranking, and geographical location (rural versus urban) for all artists and scientists' institutions. Since the program originated at Vanderbilt University/Vanderbilt University Medical Center, most artists and scientists are located at this institution. If an artist participated in multiple program years (N = 2), they were counted twice and so on. This study was approved by Vanderbilt IRB (#220157) and consent was not required.

**Table 1. Institutions of artists and scientists.** We consider minority-serving status, NIH funding ranking, and geographical location (rural versus urban) for all artist and scientist institutions. Since the program originated at Vanderbilt University/Vanderbilt University Medical Center, most artists and scientists are located at this institution. If an artist participated in multiple years of the program, then they were counted twice (N = 2).

| Group | Ranking | Institution | Students | Faculty |
|---|---|---|---|---|
| Predominantly Black Institutions | NR | Johnson & Wales University | 1 | |
| Small Liberal Arts College | NR | Bowdoin College | 1 | |
| Small Liberal Arts College | NR | Scripps College | 2 | |
| Small Liberal Arts College | NR | Sewanee: The University of the South | 1 | |
| | NR | Ohio Northern University | 1 | |
| Historically Black Colleges | 1724 | Fisk University | 1 | |
| Historically Black Colleges | 987 | Hampton University | 1 | |
| Historically Black Colleges | 988 | North Carolina Agricultural and Technical State University | 2 | |
| Asian American Native American Pacific Islander-Serving Institutions & Hispanic Serving Institutions | 407 | University of California, Merced | | 1 |
| | 301 | The University of Tennessee, Knoxville | 1 | |
| | 160 | Indiana University | 3 | |
| | 131 | University of Colorado, Anschutz Medical Campus | | 1 |
| | 115 | Tufts University | 1 | |
| | 74 | Massachusetts Institute of Technology | | 1 |
| | 72 | Medical College of Wisconsin | | 1 |
| | 49 | University of Florida | 1 | |
| | 24 | University of Wisconsin, Madison | | 1 |
| | 19 | Vanderbilt University/VUMC | 36 | 31 |
| | 14 | Washington University in St. Louis | 1 | 1 |
| | 13 | Yale University | 1 | 1 |
| | 11 | Columbia University Health Sciences | | 1 |
| | 7 | Duke University | 1 | 2 |
| | | **TOTAL** | 55 | 41 |

A.

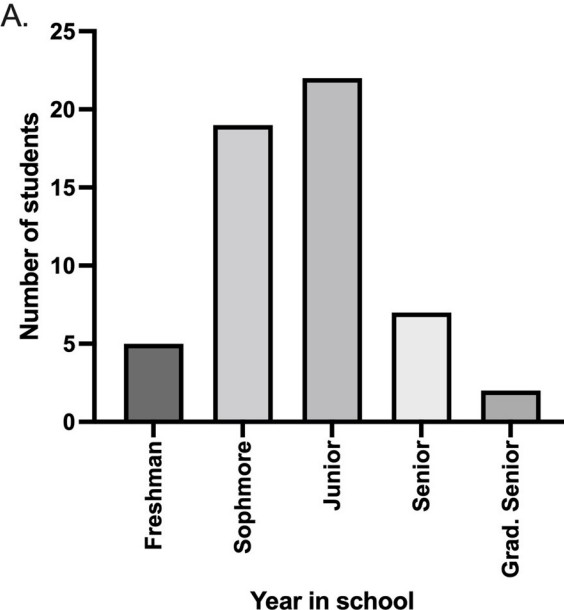

B.

|  | Science | Art |
|---|---|---|
| Self taught | 11 | 10 |
| Courses or less than 2-yrs expereince | 15 | 27 |
| Courses & greater than 2-yrs expereince | 29 | 18 |

C.

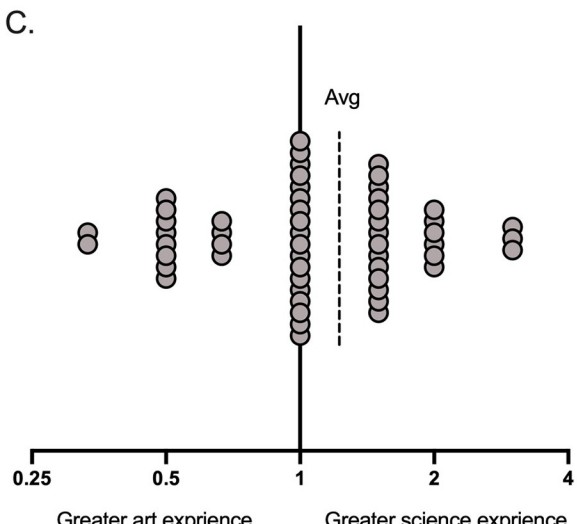

D.

|  | Major | Minor |
|---|---|---|
| Anthropology | 1 | 1 |
| Art | 2 | 6 |
| Biochemistry | 1 |  |
| Bio/Biological Sciences | 15 |  |
| Bio/Chemical & Biomolecular Engineering | 2 |  |
| Business | 1 | 2 |
| Chemistry | 4 | 4 |
| Cinema and Media Arts | 1 |  |
| Communication of Science and Technology | 4 |  |
| Computer Science | 4 |  |
| Economics | 2 |  |
| Engineering Management |  | 1 |
| Environment and sustainability studies | 2 |  |
| German Studies |  | 1 |
| History/Art History | 5 | 1 |
| Human and Organizational Development | 3 |  |
| Mathematics | 1 |  |
| Medicine, Health and Society (MHS) | 6 |  |
| Molecular & Cell Biology | 6 |  |
| Neuroscience | 7 | 1 |
| Philosophy | 1 |  |
| Physics | 1 |  |
| Political Science | 1 | 1 |
| Psychology | 4 |  |
| Religious studies | 1 |  |
| Science, Technology, and Society | 1 |  |
| Spanish | 2 |  |
| Studio Art/Fine Arts | 4 | 1 |
| Undecided | 2 |  |

**Fig 1.**

**Artists.** In total, we have had 55 artists participate in the AiR Program, which began in 2019. Artists who participated in the program were mostly undergraduates (Fig 1A). Most were either undergraduate sophomores or juniors. These student artists had a variety of majors and minors (Fig 1B). If students were pursuing multiple majors or minors, each major and minor for student was counted. The most popular majors were Biology or Biological Sciences, Medicine, Neuroscience, Health and Society, and Molecular and Cell Biology. The most common minors were art and chemistry.

**Scientists.** As we selected scientists, we were also interested in diversity, unique opportunities, and geographic location. We determined the scientists rank and found that most were full professors (Fig 2A). We also attempted to determine the scientists' schools housed within

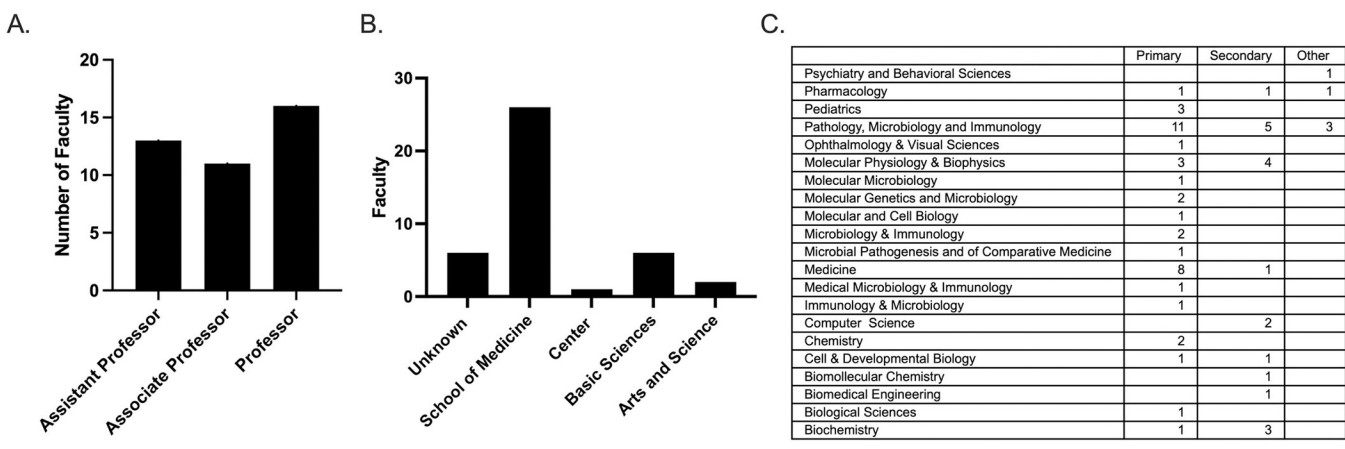

**Fig 2.**

various institutions (Fig 2B). The majority of scientists were in colleges or schools of medicine, primarily associated with medical institutions. Finally, based on appointment, we explored the scientist members' department and field of study. The majority of scientists were either in Pathology, Microbiology and Immunology, or Medicine (Fig 3C).

*Study and survey design.* This study was reviewed and approved by Vanderbilt University (Vanderbilt University IRB #220157). All participants had the opportunity not to complete the post-program survey. We have 100% of the artists complete the post-program survey and 33.3% of scientists.

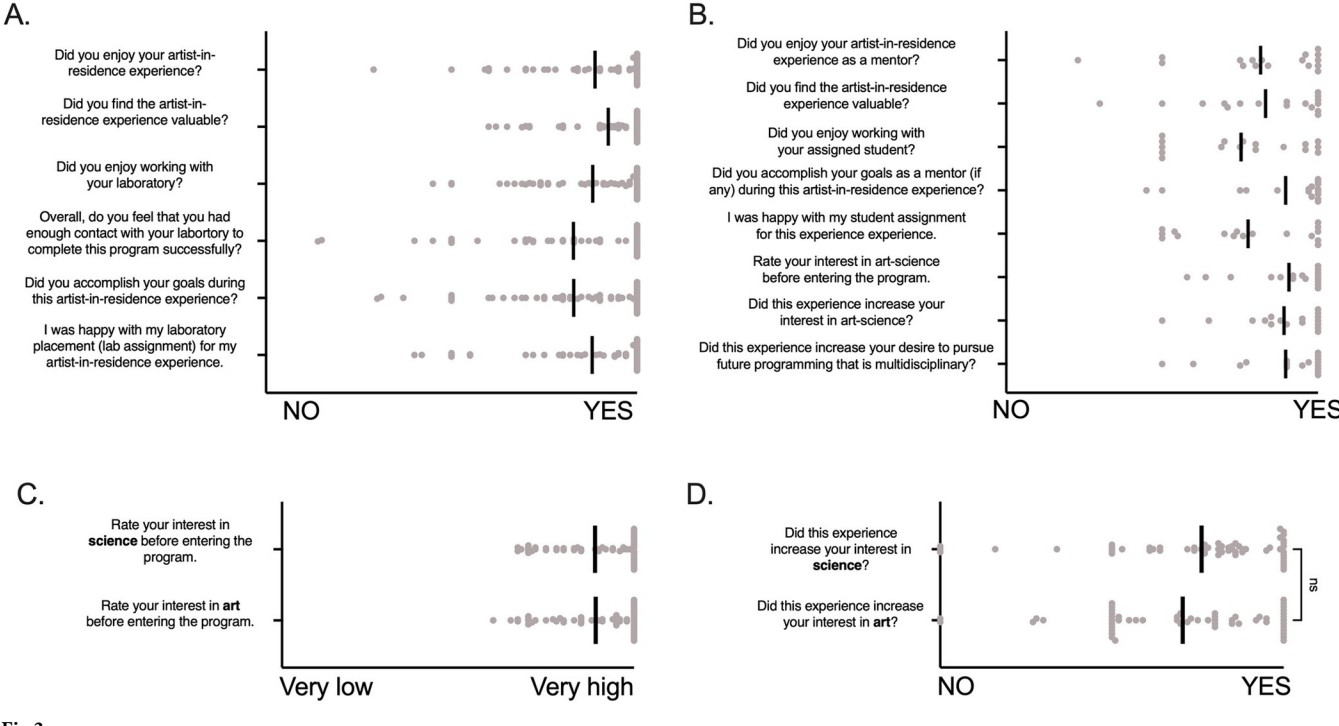

**Fig 3.**

*Quantitative analysis.* For all data graphing, we used Prism 9.0 GraphPad software [32]. Three coders were used, and each coder was asked to examine the open-ended, de-identified survey response answers independently. To measure the agreement between coders, we used the NVivo coding comparison Query Criteria for each pair of coders that reports both a weighted and unweighted Kappa score. the Kappa scores were 0.42, 0.46, and 0.46 between the three reviewers indicating acceptable agreement of coders to continue with the analysis (more information available in supplemental methods in S1 File).

**Quantitively identifying the artists experience in both art and science.** Through a qualitative analysis from artists applications, we explored and scored the experience in both art and science of the artists who participated in the program (Fig 1C). Application for the ArtLab program were shared through multiple networks. Likely the artists who applied to the program had some interest in both art and science and therefore aren't entirely random. The applications were given a score (1–3) for various experience levels in art and science. One indicated that the person was self-taught or had no experience. Two indicated that they had less than two years of practical experience (program, university-level research) experience or had only taken courses. Three indicated that they had taken courses with greater than two years of practical experience.

*Qualitative analysis.* We used NVivo [33] for all qualitative data analysis. We set out to define (1) the AiR program experience from the perspective of artist versus scientist, and (2) outcomes of the program for artists versus scientist. We performed a constructive thematic analysis of open-ended survey responses and coded open-ended responses into themes to answer these questions. The themes were informed by the program facilitators' own experience and scientists' responses. From this perspective and bias the theme generated about potential concerns related to "Interaction with program organizers," "Responsiveness of program organizers when needed," "Communication with the artist," and "Time to complete the project." We combined the open-ended survey responses of artists and scientists from a variety of prompts, including "Please explain your answer," "What was your favorite aspect of the artist-in-residence experience?" "What was your least favorite aspect of the artist-in-residence experience?" and "What could make this experience more valuable in the future?" The author reviewed all open-ended responses to the post-program survey. Not all responses were required to be coded by the coders if they did not fit the themes. For more information, please see the supplementary methods in S1 File.

## Results

### Incoming experience levels of artists

Overall, many of the artists took courses and had more than 2-years of university-level research experience in science and some courses or less than 2-years of program experience in art. To represent the balance of science versus art training, we examined the ratio of science over artist training (Fig 1D). A value of 0.33 indicated much greater artistic experience over science experience, 1 indicating an equal amount of artistic and scientific training, and 3 indicating much greater science experience than art experience. The mean value was 1.224 indicating greater scientific experience to artistic experience general for the artists.

### Experience in the program

Both artists and scientists were asked a series of questions about their experience in the program. The answers were collected on a sliding scale from 1–100 with (0 meaning "NO" or "Very Low" equaling and 100 meaning "YES" or "Very High" depending on the question). Each response is shown as an individual point, and the mean response is indicated with a

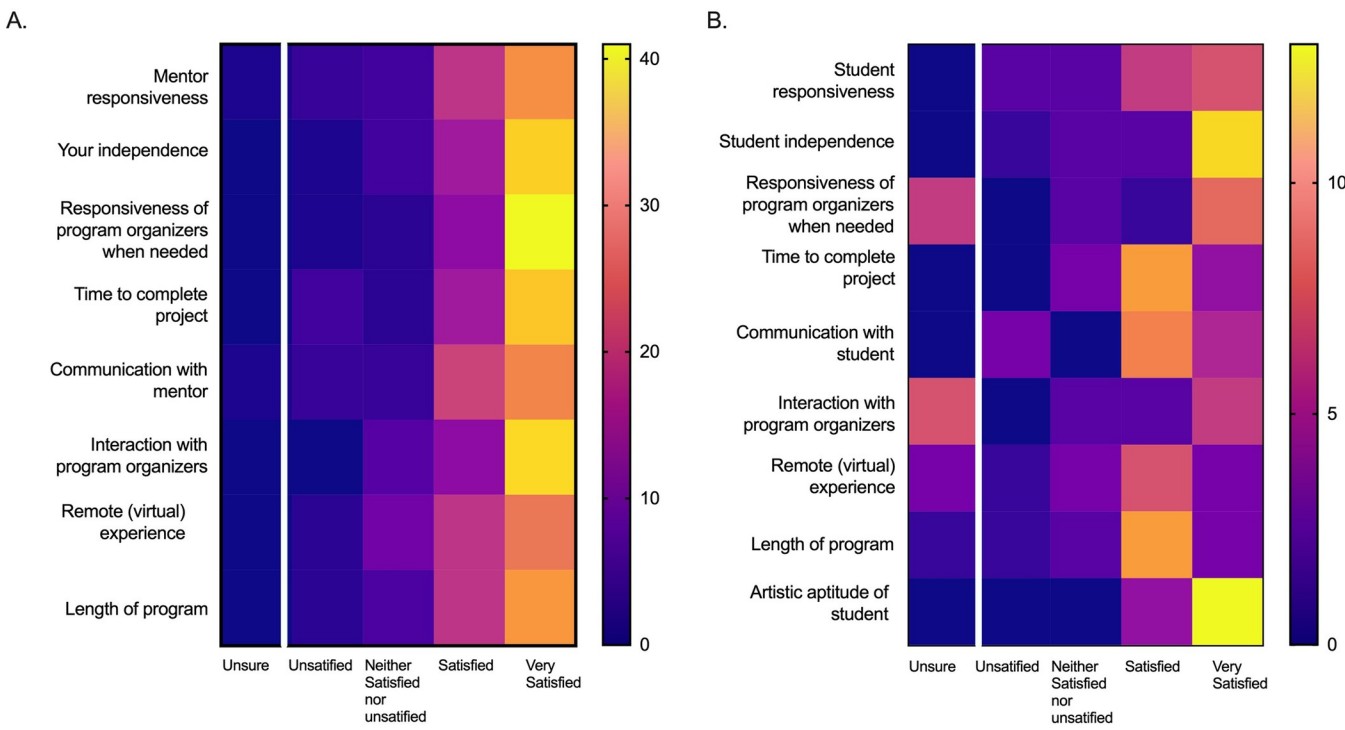

**Fig 4.**

black bar. Overall, both artists and scientists overwhelming responded "YES" to all questions posed (Fig 3A and 3B). Artists were also asked the rate their interest in science and art before entering the program (Fig 3C). Artists responded to being very interested in both art and science equally. When asked if participating in the program increased their experience, artists overall answered "YES" to the program increasing their interest in science and art (Fig 3D).

Within the survey, we also examined Likert scale data from the final program survey that explored artist and scientist satisfaction with various aspects of the program (Fig 4). This number of responses for each prompt is reported via a heat map. Overall, artists reported being "very satisfied" with all provided prompts listed in Fig 4. Scientist responses were more varied, particularly regarding "The remote (virtual) experience," "Interaction with program organizers," "Responsiveness of program organizers when needed," "Communication with the artist," and "Time to complete the project." We used these prompts to inform a constructive thematic qualitative analysis conducted on the open-ended survey questions from both artists and scientists.

## Word clouds as an overview of open-ended responses

This led us to explore a deeper understanding of the artist versus scientist experience. We, therefore, collected all open-ended responses from both artists and scientists for analysis. Before beginning a directed content analysis, we generated a word cloud of open-ended survey responses from artists and scientists (Fig 5). Word clouds are used in various contexts as a means to provide an overview by distilling text down to those words that appear with highest frequency. The top 100 words with a minimum length of 3 letters were plots in a word cloud. All responses from artists (A) and scientists (B) were coded and grouped based on either being from an artist or a scientist participant. In viewing the word clouds, an overview of the words

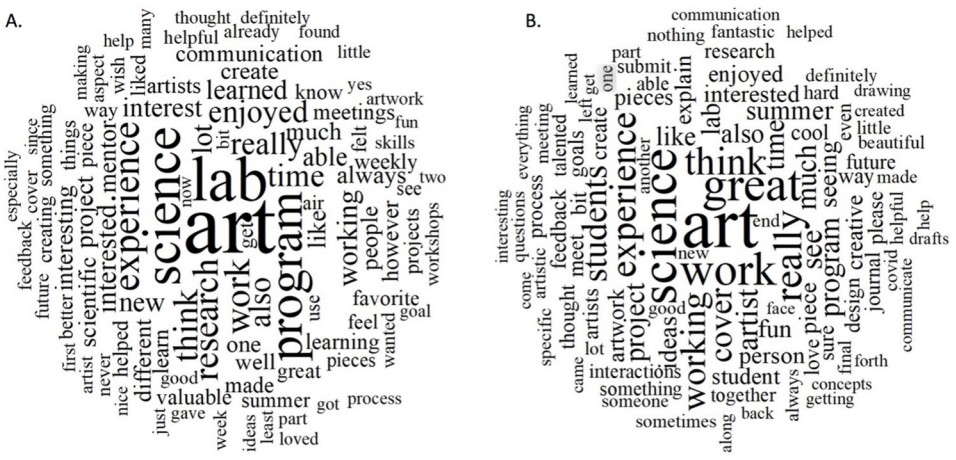

**Fig 5.**

that occur most often within the text is visualized. These words included things like art, science, cover, enjoyed, interested, communication, people, etc.

## Informed thematic analysis for lower-performing satisfaction areas among scientists

Based on the satisfaction analysis from Fig 4, we wanted to better understand the areas that did not exceed expectations. The four themes investigated as part of the program experience included communication between program participants, program design and implementation, time or length, and the virtual or remote nature of the program. These themes were informed based on results from Fig 4. We compared the number of responses as well as the amount of coverage each theme had among all responses, or percent coverage, to minimize the disparity between the number of responses between artists and scientists. All percentages below refer to the percentage of all open-ended text responses from scientists or artists that represents that theme. Low percentages indicate that there are fewer comments about that theme, while higher percentages indicate that there was more comment about that theme."

**Communication between program participants.** The first area we explored with the thematic analysis was communication between program participants (Fig 6A). We flagged any comment related to program communication between artists, scientists, or artist-scientist partners. Coders were instructed not to include comments on science communication for a public audience to better understand the communication between the artist and scientist partnership. Overall, there were 318 coding references for artists (20.96%, coverage of all open response text from artists) and 85 coding references for scientists (28.93%, coverage of all open response text from scientists) related to communication (S4 and S5 Figs). There was a balance of both positive and negative codes from both artist and scientists about communication between participants but no significant difference (S1 Fig).

For artist responses, 168 coding references were positive (10.83%), and 150 were negative (10.44%). Examples of positive artist codes include, "My discussions with my mentor have been very efficient," "I feel that my contact person has been perfect with communicating," and "I thoroughly enjoyed being paired with a lab and our weekly meetings to see what the other artists were up to." Some examples of negative artist comments include, "I wish I had more communication with my mentor," "Besides the lab I was working with, I didn't have any people to ask questions because I had never done a residency before, so it was feeling around in

## Communication between participants

## Program design and implementation

## Time or length of the program

## Virtual or remote nature of experience

**Fig 6.**

the dark to find something that worked," and "Discussing and deciding on the subject of the piece pushed the creation of the piece back significantly, and I believe the execution of my final piece suffered because of it."

For scientists, 44 coding references were positive (17.04%), and 41 were negative (12.94%). Examples of positive scientist responses include, "We got along great, and I think it came through in the output," "She did a phenomenal job of integrating scientific paper abstracts into art ideas and went above and beyond in soliciting feedback and generating drafts," and "She provided updates regularly and responded well to suggestions. She also was eager to learn about this project and asked questions when needed." Some examples of negative scientist responses related to communication include, "Aside from our initial interaction by phone, there was not much follow-up or dialogue," "It was sometimes hard to schedule meetings with the artists," and "Trying to schedule with the artists, it seems like they had many other demands on their time."

**Program design and implementation.** The second area that we examined relating to the program experience was the program's design and implementation (Fig 6B). This included comments that relate to program organizers, program organization, including learning goals, objectives, emails, approach, guidance, and other organizational elements. Overall, there were 334 coding references for artists (24.47%) and 80 coding references for scientists (23.44%) related to the program design and implementation. There was a balance of positive and

negative codes from both artists and scientists about program design and implementation but no significant difference.

For artists, 167 coded references were positive (13.74%), and 167 coded references were negative (10.95%). Examples of positive comments include, "Filling the survey provided by _____ also helped me clear my thoughts and inspired me a lot," "having the opportunity to have my artwork printed and included in an exhibition for the year is incredibly rewarding and something that I hope to continue in the future," "I thoroughly enjoyed being paired with a lab and our weekly meetings to see what the other artists were up to." Some examples of negative comments related to program design and implementation include, "I just wish the beginning of it was more coordinated. Besides the lab I was working with, I didn't have any people to ask questions because I had never done a residency before, so it was feeling around in the dark to find something that worked," "However, I wish there was a clear publication or project selected from the beginning of the project. Discussing and deciding on the subject of the piece pushed the creation of the piece back significantly, and I believe the execution of my final piece suffered because of it," and "I was also expecting for there to be more technical work-shops learning techniques in software."

For scientists, 46 coded references were positive (12.68%), and 34 coded references were negative (10.76%). Some examples of positive comments from scientists include, "the best part was seeing our work from an artistic viewpoint and developing an abstract drawing that would present our work in a creative and aesthetic way," "I made sure to engage multiple lab members in the process to make the experience as rich as possible for everybody," and "I most enjoyed the brainstorming and concept sessions." Some examples of negative comments related to program design and implementation include, "having a little bit of structure to go back and forth on ideas and drafts could have been helpful," "artist not seeming to have full guidance and direction," and "having a better understanding of the expectations might have helped me . . ., for example, if I knew that it was expected to go back and forth 4–6 times on drafts then I would know that giving feedback five times is reasonable and not too much to ask."

**Time or length of the program.** Next, we exampled the theme of time or length of the program (Fig 6C). This theme related to all comments that mention the scheduling, timing, not finishing a project due to time limitations, time management, etc. Overall, 116 coded references for artists (7.39%) and 22 coding references for scientists (4.56%) related to the program time and/or length. There were more negative codes from both artists and scientists about communication between participants but no significant difference between artists and scientists in terms of positive or negative codes.

For artists, 13 coded references were positive (0.77%), and 103 coded (6.65%) references were negative. Examples of positive comments include, "I enjoyed the way of building up a set of works from continuous efforts in a given amount of time." Some examples of negative comments include, "Overall I enjoyed the program, but feel that it was too short to do my best work," "This would have been an incredibly rewarding experience; my only regret is taking on too much this summer, so I did not have the time to devote to this program," and ". . . the stress of completing the weekly surveys and watching lectures made everything seem rushed."

For scientists, four coded references were positive (0.68%), and 18 coded references were negative (3.95%). Some examples of positive comments include, "It was great practice in scientific communication for me, and didn't take up too much time," and "it was great to do this over the summer." Some examples of negative comments related to the length of the program include, "[my least favorite part] How short it was," and "Time flew by quickly summer, and I wish we could have had a bit more time to work together."

**Virtual or remote nature of experience.** Finally, the last theme that we examined as a part of the program experience was the virtual or remote nature of the program (Fig 6D).

Comments coded to this theme included any mention of being in-person, working remotely, or interacting virtually. Overall, there were 61 coded references for artists (2.76%) and 42 coded references for scientists (8.53%) related to the program's virtual or remote nature. There were more negative codes associated with the virtual or remote nature of the experience from both artist and scientists about communication between participants but no significant difference. However, it was trending towards more negative scientists' responses than artists.

For artists, ten coding references were positive (0.64%), and 51 coded (2.71%) references were negative. An example of a positive artist response was, "I thought the program was well adapted to a remote-only environment." An example of a negative comment about the virtual experience included, "The only thing that influences my experience is the technical inconvenience of communication sometimes due to internet control in China," "However, being remote has made it difficult to find inspirations for the project," and "However, I think that an in-person experience in the future would be even better!"

For scientists, two coding references were positive (0.38%), and 40 coding references were negative (8.33%). An example of positive comment about the virtual experience was, "But overall, it was a fantastic experience, and converting to remote still worked great!" Some examples of negative comments related to the virtual experience include, "I think it was a bit hard to do this remotely, but I can understand why it was necessary," "The fact that most of it was virtual [was my least favorite part]," and "[my least favorite part was] not having her nearby or getting to meet her in person before the program started."

## Thematic analysis of benefits for artist-scientist partnership

In addition to running a thematic analysis on the program experience, we performed a parallel investigation to explore program outcomes. The four themes related to program outcomes that were coded include mentoring experience, way of thinking, learning something new, and science communication to the public or general audience. Again, we compared the percent coverage of each theme to minimize the disparity between the number of responses between artists and scientists.

**Mentoring experience.** The first theme coded for part of the program outcomes was the mentoring experience (Fig 7A). This included comments related to either a artist or a scientists mentoring situation in a programmatic sense. This did not include specific statements about mentoring styles but focused more on mentorship. Overall, there were 339 coding references for artists (22.5%) and 151 coding references for scientists (42.49%) related to the mentoring experience. There were more positive codes from both artists and scientists about the mentoring experience. There was no significant difference between artists and scientists regarding positive or negative codes.

For artists, 275 coded references were positive (15.51%), and 64 coded references were negative (7.1%). Some examples of positive comments are, "My lab was incredibly enthusiastic and welcoming," "I enjoyed working in conjunction with the ___ lab to explore where my two interests merge," and "I enjoyed getting to know ____ and ____." Some examples of negative comments related to mentoring included, "The only thing keeping me from loving this experience 100% was the absence of my lab PI throughout my time in the program," and "I felt that it was sometimes unclear how the PI wanted to use my work and what the goals were for a final product."

For scientists, 127 coded references were positive (32.39%), and 24 coded references were negative (10.5%). Some examples of positive comments include, "It was a blast working on art projects with such a creative and kind person," "___ came in with so many ideas and asked really insightful questions about the science to represent everything correctly," and "____ and

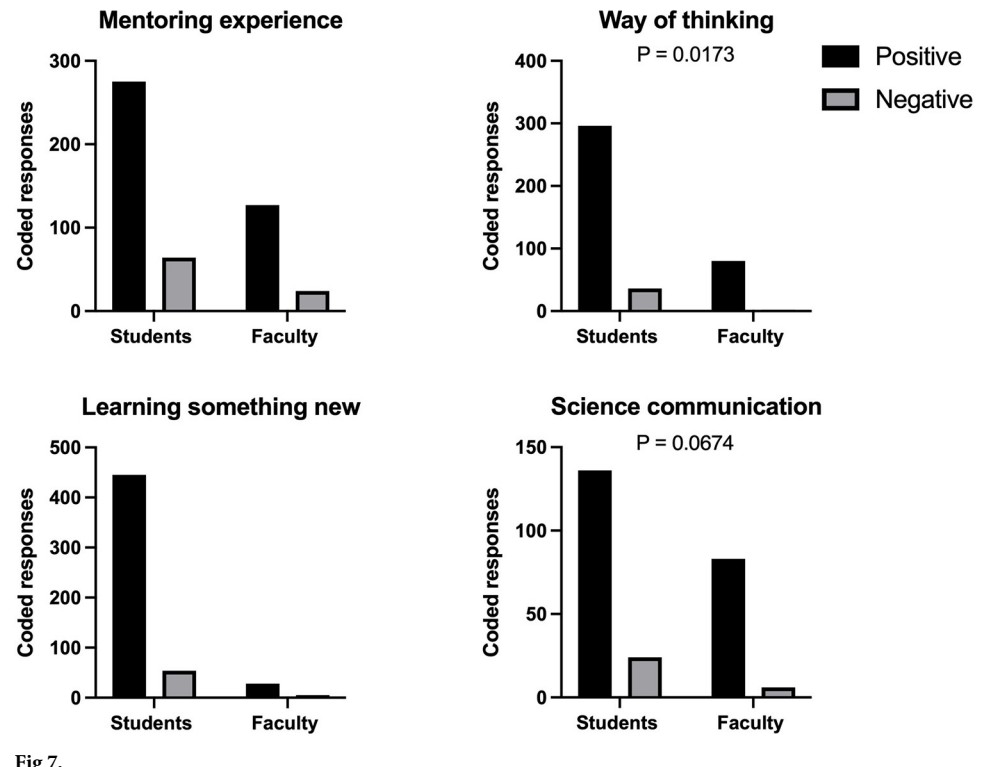

**Fig 7.**

I built a really positive rapport that went beyond our specific project. I felt like it was a really great mentoring experience, and we plan to keep in touch!" Some examples of negative comments related to mentoring included, "Aside from our initial interaction by phone, there was not much follow-up or dialogue," and "I only partially mentored the artist. I was more of a liaison between her and the graduate artists and postdocs."

**Way of thinking.** The second theme coded for was "way of thinking," which was used to indicate comments related to how the experience led to a change in the artist or scientist members' way of thinking related to science, art, or in general (Fig 7B). Overall, there were 332 coded references for artists (23.48%) and 82 coded references for scientists (25.05%) related to a change in thinking. There were more positive codes from both artists and scientists about the theme and there was a significant difference between artist and scientists. Overall, scientists had a more significant percentage of positive codes for the theme and fewer negative codes than artists.

For artists, 296 coding references were positive (20.56%), and 36 coding references were negative (3.03%). Some examples of positive comments related to the way of thinking included, "I enjoyed exploring the art side of science," "I really think that I was encouraged to explore and try new things in the program, and I've learned many new art techniques," and "it was challenging but fun to figure out how to represent them visually." An example of a negative comment relating to the "way of thinking" was, "Thinking up creative ways to communicate research accurately is a difficult task, and I would have to sacrifice either creativity or accuracy or ease of understanding."

For scientists, 80 coding references were positive (24.44%), and two coding references were negative (0.61%). Some examples of positive comments related to 'way of thinking" are, "It was great to think creatively with a true artist about visual scientific communication," "these are

very talented artists that introduced new mediums and ideas of how to communicate science to others outside of my circle," and "It was fun to explain the project that I've been working on to someone with no science background, and then to see it brought to life by the artist." An example of a negative comment related to "way of thinking" was, "I already value and search for [the intersection of art and science] in our research efforts."

**Learning something new.**   The next theme that we explored as a part of program outcomes was "learning something new" (Fig 7C). We used this code to tag any response where the individual indicated that they had learned or experienced something new or unique related to art or science from the program. Overall, there were 499 coded references for artists (27.58%) and 33 for scientists (10.73%) related to learning something new. There were more positive codes from both artists and scientists about "learning something new," and there was no significant difference between artist and scientists.

For artists, 445 coded references were positive (23.23%), and 54 coded references were negative (4.53%). Some examples of positive comments related to learning something new included, "It was interesting because I learned more about the _____ lab's work by reading the manuscripts while looking up terms on my own," "I really think that I was encouraged to explore and try new things in the program, and I've learned many new art techniques," and "Through the experience in AiR program, [I] not only got to know biological knowledge and research works but also learned so many new art techniques." Some examples of negative comments related to learning something new included, "I wish I had learned more overall," and "The workshop meetings sometimes felt very separate from what we were doing in the program, and I sometimes struggled to make a connection to my own work."

For scientists, 28 coded references were positive (9.82%), and five coded references were negative (2.14%). Some examples of positive comments from scientists about learning something new included, "it was interesting to hear an artistic perspective on science," "[My favorite part was] finding a way to relate to someone who is not in my field," and "Having that creative vision and artistic skill to produce cover art is something I'd never be able to do on my own." An example of a negative comment related to learning something new was, "My interest [in art-science] was already high, it didn't increase." Comments like these often corresponded to those with some pre-existing art-science background.

**Science communication to the public or general audience.**   Finally, we explored the theme of "science communication to the public or general audience" (Fig 7D). We used this code to tag any comment or response that relates to a positive impact that the program made on the artist or scientists members' ability to communicate or understand science. Overall, there were 160 coded references for artists (13.71% coverage) and 89 coding references for scientists (26.22% coverage) related to science communication. There were more positive codes from both artists and scientists about the "science communication to the public or general audience," and there was a trend towards the difference between artist and scientists. Overall, scientists had a greater percentage of positive codes for "science communication to the public or general audience" and fewer negative codes than artists.

For artists, 136 coding references were positive (11.85%), and 24 coding references were negative (1.9%). Some examples of comments related to positive program outcomes around the theme of science communication included, "I loved exploring scientific communication and illustration" and "I liked working with a lab and figuring out how to communicate the scientific concepts in a creative way." An example of a comment negatively related to science communication was, "I thought we would learn more about how to effectively communicate science through art, which we may have briefly touched on in some sessions but not completely".

For scientists, 83 coding references were positive (24.1%), and six coding references were negative (2.11%). Some examples of positive comments from scientists that were related to science communication included, "It was great to think creatively with a true artist about visual scientific communication," "Working with _____ to design cover art was a fun and incredibly valuable experience," and "This was a great experience that resulted in a beautiful piece of art that encompassed the major concepts of our research." An example of negative comments related to science communication was, "It was fun, and I am excited to have this art piece, but whether we'll end up using it professionally (journal cover, etc.) I'm not so sure."

## Discussion

The goal of this paper is to review post-survey data from the VI4 AiR program to (1) determine what the benefits are for both scientists and artists, (2) identity ways to improve the design and implementation of the program, and (3) develop best practices for art-science based partnerships. Our findings suggest an opportunity for alterations within the program to enhance the program experience and that the program experience was similar between artists and scientists. Overall, program outcomes were positive by all participants, which suggests the benefits in participating in the AiR program includes experiencing mentorship, altering one's way of thinking, learning new skills, and implementing science communication. Our programmatic evaluation is particularly relevant for undergraduate artists and scientist in the biomedical sciences. However, our results suggest that particularly for "way of thinking" and "implementing science communication," scientists may benefit more.

As we began to explore our data, we became interested in defining the program experience from artist versus scientists. In many cases, both artists and scientists have similar positive and negative experiences during the artist in residence program. For instance, there was a significantly higher number of negative responses regarding the time or length of the program and the virtual and remote than positive responses from both artists and scientists. But not all our questions had high negative response rates, and therefore we do not believe that the larger number of negative responses for themes like "time or length of the program" and "virtual or remote nature of the experience" is simply a survey error. These findings suggest the need to modify the program's virtual nature, whether that be shifting to a hybrid or fully-in person experience. If the program were to remain virtual, then additional effort should be put on demonstrating the benefits of this modality, particularly for scientists. Altering the length of the program should also be addressed to better the overall experience for both artists and scientists.

As we reviewed the open responses, we were also interested in better defining artists and scientist benefits. We created a thematic analysis based on the open-ended responses related to mentoring, way of thinking, learning new skills, and science communication. In many instances, the scientists and artists have similar values of the program outcomes. For example, there was a significantly higher percentage of positive responses regarding "mentorship experience' and "learning something new" than negative responses from both artists and scientists. However, some artist and scientist perspectives on program benefits differed. For instance, artist participants were more positive about "learning something new from the program" than scientist. One of the most surprising aspects of our analysis suggests that "way of thinking" and "science communication to the public or general audience," is indicated as a more significant beneficial gains for scientists as compared to artists. This information could be useful in future designs of art-science programs by customizing the programming for these distinct benefits based on group.

From this analysis future program development and targeted outcomes can be improved. This includes specific modifications to the program, including a hybrid format, with options for in-person components. This modification could give more insight into an optimal way of collaborating versus restricting to virtual or in-person. Based on responses, lengthening the program, or putting less demand on deadline for the final product could increase positive experiences. However, there could also be concerns about not specifying program outcome goals and creating deadlines. Responses regarding communication suggest subjectivity based on the artist-scientists matching and could be improved with implementing more communication methods and increased collaboration with program directors to ensure efficient communication. Responses for program design also request more clarification of the overall process of the program and final product expectations.

There are several limitations to this study that readers must take into consideration. First, as previously mentioned, the themes developed for the constructive thematic analysis were based on the program directors' first-hand experience. Therefore, the themes might introduce confirmation bias based on the program director's experience and subjective perspective. Because we used these defining themes to guide the data analysis, there may be more or parallel information within the data set that was not collected or represented in the final analysis. We attempted to mitigate this bias by using three coders. Another major limitation of this study was selection bias because the artists and scientists that participated in this program have a pre-established interest in merging art and science. This is particularly evident in Fig 3, where artists were asked the rate their interest in science and art before and after entering the program. Because the artist interest was extremely high before entering the program, we reached a ceiling effect where there was less than expected growth in interest after participation. Finally, it is unclear why scientist participation was only 33% for the post-program survey. However, we feel that these responses represent the overall pool of scientist participants. Future studies conducted by those outside of the program and involving artist-scientist pairs that do not have pre-established interest in art and science intersection may expand the impact of this study. One final consideration is that the "artists" in this program are mostly undergraduate students, and they are working with faculty-level scientists and their teams including graduate and post-doctoral-level people. It is important to consider the power dynamics intrinsic to this relationship. We acknowledge that the collaboration between the artists and scientists may be skewed due to this power differential.

Overall, this analysis summarizes the AiR experience from the perspective of both artists and scientists to understand how to better design the program and benefit both artists and scientists. Program outcomes of "mentorship experience," "way of thinking," "learning something new," and "science communication" were beneficial among both artists and scientists. Still, our analysis suggests that for "way of thinking" and "science communication," there could be even more significant gains for scientists as compared to artists. There is a need to improve the skills and provide training opportunities for scientists in visual science communication. The AiR experience might be one avenue for science communication training opportunities that also provides a tangible visual science communication product for the scientist. From this analysis, we can take clear steps to maximize the program experience, and likely enhance the program. For instance, comments regarding the mentorship experience included communication which we can address by adjusting the program design. Future work should consider measuring specific skills such as how participation in the program impacts creativity, objectivity, perseverance, spatial reasoning, and observational acuity. However, this work begins to define promising benefits of art-science programs, diagnoses of difficulties such programs, provides insights on design and implementation, and future directions to support art-science partnerships.

## Supporting information

**S1 File. Coding guide.** Available on Qualitative Data Repository (https://data.qdr.syr.edu/dataset.xhtml?persistentId=doi:10.5064/F6SUSRIC).
(DOCX)

**S1 Fig. Comparing student to faculty responses.** To determine if students and faculty responded differently to the prompts about their experience in the program, we graphed student and faculty responses identical to Fig 3 on the same graph. There was no significant difference between faculty and staff responses to survey questions.
(PNG)

**S2 Fig. Coding themes map.**
(PNG)

**S3 Fig. Screen shot of inter-coder reliability Kappa scores between reviewer 1 and 2 from Nvivo.** Available on Qualitative Data Repository (https://data.qdr.syr.edu/dataset.xhtml?persistentId=doi:10.5064/F6SUSRIC).
(PNG)

**S4 Fig. Screen shot of inter-coder reliability Kappa scores between reviewer 3 and 2 from Nvivo.** The NVivo files can be downloaded from the Qualitative Data Repository (https://data.qdr.syr.edu/dataset.xhtml?persistentId=doi:10.5064/F6SUSRIC).
(PNG)

**S5 Fig. Screen shot of inter-coder reliability Kappa scores between reviewer 1 and 3 from Nvivo.** The NVivo files can be downloaded from the Qualitative Data Repository (https://data.qdr.syr.edu/dataset.xhtml?persistentId=doi:10.5064/F6SUSRIC).
(PNG)

**S6 Fig. Screen shot of NVivo program showing the program evaluation qualitative data.** The NVivo files can be downloaded from the Qualitative Data Repository (https://data.qdr.syr.edu/dataset.xhtml?persistentId=doi:10.5064/F6SUSRIC).
(PNG)

**S7 Fig. Screen shot of NVivo program showing the program outcomes qualitative data.** The NVivo files can be downloaded from the Qualitative Data Repository (https://data.qdr.syr.edu/dataset.xhtml?persistentId=doi:10.5064/F6SUSRIC).
(PNG)

## Acknowledgments

We thank the scientists and 55 artists that participated in our programming.

## Author Contributions

**Conceptualization:** Max E. Joffe, Eric P. Skaar, Kendra H. Oliver.

**Data curation:** Qi (Kathy) Liu, Kendra H. Oliver.

**Formal analysis:** Skylar Cuevas, Qi (Kathy) Liu, Helen Qian, Kendra H. Oliver.

**Investigation:** Eric P. Skaar, Kendra H. Oliver.

**Methodology:** Skylar Cuevas, Helen Qian, Kendra H. Oliver.

**Project administration:** Karisa Calvitti, Megan Schladt, Kendra H. Oliver.

**Resources:** Karisa Calvitti, Kendra H. Oliver.

**Software:** Helen Qian, Kendra H. Oliver.

**Supervision:** Eric P. Skaar, Kendra H. Oliver.

**Validation:** Kendra H. Oliver.

**Visualization:** Kendra H. Oliver.

**Writing – original draft:** Skylar Cuevas, Max E. Joffe, Eric P. Skaar, Kendra H. Oliver.

**Writing – review & editing:** Skylar Cuevas, Helen Qian, Max E. Joffe, Karisa Calvitti, Eric P. Skaar, Kendra H. Oliver.

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
