## [Decision Letter · Decision Letter 0]

25 Apr 2022

PONE-D-22-04237Differences in self-reported benefits for student-artist versus faculty experiences in a virtual artist-in-residence program.PLOS ONE

Dear Dr. Oliver,

Thank you for submitting your manuscript to PLOS ONE. After careful consideration, we feel that it has merit but does not fully meet PLOS ONE’s publication criteria as it currently stands. Therefore, we invite you to submit a revised version of the manuscript that addresses the points raised during the review process.

We look forward to receiving your revised manuscript.

Kind regards,

Professor Dr. Mehmet Serkan Kirgiz

Academic Editor

PLOS ONE

Journal Requirements:

2. Please include your tables as part of your main manuscript and remove the individual files. Please note that supplementary tables (should remain/ be uploaded) as separate "supporting information" files.’

3. Please ensure that you refer to Figure 7 in your text as, if accepted, production will need this reference to link the reader to the figure.

6. Thank you for stating the following in the Funding Section of your manuscript:

“This program is supported by the Vanderbilt Institute for Infection, Immunology, and Inflammation (VI4), Burroughs Wellcome Fund, The Wond'ry Center for Innovation, The Curb Center for Art, Enterprise & Public Policy, and The Communication of Science and Technology program within the College of Arts and Sciences.”

We note that you have provided additional information within the Funding Section that is not currently declared in your Funding Statement. Please note that funding information should not appear in the Acknowledgments section or other areas of your manuscript. We will only publish funding information present in the Funding Statement section of the online submission form.

Reviewers' comments:

Reviewer's Responses to Questions

**Comments to the Author**

1. Is the manuscript technically sound, and do the data support the conclusions?

Reviewer #1: Yes

2. Has the statistical analysis been performed appropriately and rigorously? 

Reviewer #1: Yes

3. Have the authors made all data underlying the findings in their manuscript fully available?

Reviewer #1: Yes

4. Is the manuscript presented in an intelligible fashion and written in standard English?

Reviewer #1: Yes

5. Review Comments to the Author

Reviewer #1: The paper sets out to better understand programmes that bridge the arts and sciences. The paper focuses on the results of such collaborations over three summers by surveying students and academic staff. The data for the publication is based on survey data, with a particular focus on “positive” and “negative” experiences. The survey results have been extensively analysed and areas of strength and that could do with improvement are highlighted. The paper offers suggestions about how to improve future art/science collaborations.

The claims appear to be properly placed in the context of the previous literature and for the most the authors treated the literature fairly. However, there are instances were some more citations would be useful. For example, the first line of the introduction (line 55-6) makes a statement that could do with a citation to support it.

The authors show a clear understanding of how to analyse surveys, referring to concepts and literature in a knowledgeable manner. They have provided very detailed analyses and good charts. However, there are instances that need minor clarifications. For example, percentages are given that do not appear to clearly link to an overall top-level figure. This should be cleared up with a line or two of text, possibly when the first occurrence of the numbers appears. (For example, see line 244, 245, 248 and so on). Otherwise, if the basis of these number is more complicated, a (supplementary information) table could help to summarise the information. Or if this is already in the supplementary information, it would be good to more explicitly state this. (Perhaps it can be inferred from somewhere, but I could not do so.)

As for the literature, the authors have a contemporary and convincing understanding of the topic of art and science collaboration. The literature cited also indicates they have a good understanding of survey methodologies and good practice in this domain.

The data and analyses support the authors’ claims for the most, however they should more extensively address possible biases that could exist in their survey results.

• It would be good to add a line or two discussing the possible limitations of their analysis. I.e. to simply acknowledge that the results specifically represent those who participated, and might not reflect that of the entire population of potential participants. It would be useful to briefly refer to some literature that discusses how views might differ from those that did not participate.

• Details of the methodology give a good idea how to reproduce the experiments, this is further supported by the willingness of the authors to share information, as they have indicated they are willing to do.

• The manuscript well organized and written clearly enough to be accessible to non-specialists, some minor revisions are needed though. More specifically, the abstract, introduction, methodology are clear and well explained. However, the results section is perhaps too long and descriptive. The language is repetitive at times and could do with some rewriting (which should not be a major task, considering the large amount of work that appears to have gone into this analysis). A solution might be to move some of this material to the supplementary documentation. The information is often important, but the repetitive and descriptive nature of it is tough for a reader and distracts from the more interesting results that the authors have uncovered.

To reiterate, this paper is a good and valuable piece of work, even if what follows seems long. What follows is a list of minor points. The authors can disagree or disregard some of these if they wish, but it would help to address as many as they reasonably can.

• Line 49. If you like, you could briefly mention a strong example here.

• Lines 55-6. A citation would be good.

• Line 58. Possibly a word missing.

• Line 65-6. Check sentence.

• Line 66-7. Explain a little what is meant by “recipe”.

• Line 76-8. This sentence could be moved a bit earlier. Also add a citation for this, or clarity that it is the authors’ definition/view, unless I misunderstand and the statement is trivial in some way.

• Line 96. “K-12” is not clear to an international audience, so it would be good to explain this.

• Line 100. This could use a citation, unless the authors’ want to make it clear that this is in their view.

• Line 116. The “n=2” could give the impression that there are no examples of people who’ve been included more than once or twice. So maybe add a few words to make it clearer… maybe simply add “and so on” to the end of the sentence.

• Line 126. Does the representation in Fig 2A tend to be what is the usual distribution in a university? If this is hard to answer, then there is no need to do so.

• Line 131. I’m unsure what the number represents here. I think it’s a grant number, but it’s unclear.

• Line 133. As earlier discussed, it would be interesting to mention a little about who didn’t participate, and why? Or something about how you feel this represents the overall pool of possible participants. Or acknowledge that you can’t do that for sure.

• Line 134. The bold text doesn’t appear to be the correct name?

• Line 134. Provide a link/citation for the software.

• Line 136-7. Mention a bit more on how the sample was selected. To make it more convincing that it was reasonably “random”.

• Line 142. Add link/citation for the software.

• Line 145. Make it clearer what questions you are referring to. The previous “general queries” are not stated as questions.

• Line 146-7. Is there anything interesting to be mentioned about biases that might be introduced by the themes being defined (or informed) by the programme facilitators experiences?

• Line 152-3. What is the significance (if any) of the lack of need for all responses to be coded?

• Line 186. Is there much of a degree of subjectivity in what they define as positive and negative? This is perhaps dealt with in what you say next, about having multiple coders and the Kappa score?

• Line 192-3. As the Kappa score can be negative, can you also mention what are the significance of negative scores?

• Line 194. Mention where in the supplementary material. I.e. be more specific.

• Line 197. Citation/link for the software (if not already there).

• Line 213-4. It’s not clear what a “yes” is (unless I’ve missed something). I assume it’s an average over 50%? Briefly state this if so. But there are a few that are less than 50%, so clarify what is going on here.

• Line 221. Might be worth defining what is meant by “prompt”, if not already done.

• Line 229. The word clouds are nice. But I don’t think you say much about them. One or two sentences could work well on it here. The authors have generated good material here, so I think it would be nice to make more of it.

• Line 232. I’m not sure what “up-biased” means.

• Line 235. Figure 4?

• Line 236. It could be clearer to mention why four themes were selected.

• Line 243-4. Is this because the focus is meant to be between student and faculty? Maybe make this clearer.

• Line 301. The coverage numbers are low here. Is this significant in some way?

• Line 447-9. Is it possible that this larger number of negative responses… is this “simply” a common feature of survey results? That more people tend to show negative views?

• Lines 320-1. Is there significance of the more negative trending in faculty responses than students?

• Lines 347-8. Is there significance of this?

• Line 370. This is an example of opportunity rewrite/improve the text. It might not be necessary to have “way of thinking” in quotes in the paragraph three times.

That is all. Thank you for an enjoyable read.

6. PLOS authors have the option to publish the peer review history of their article (what does this mean?). If published, this will include your full peer review and any attached files.

Reviewer #1: No

---

## [Author Response · Author response to Decision Letter 0]

6 Jun 2022

Thank you very much for the thorough and thoughtful review of our manuscript. Your comments were very insightful and improved the manuscript's clarity and impact. 

Best wishes, 

Kendra

---

## [Decision Letter · Decision Letter 1]

8 Aug 2022

PONE-D-22-04237R1Differences in self-reported benefits for student-artist versus faculty experiences in a virtual artist-in-residence program.PLOS ONE

Dear Dr. Oliver,

Thank you for submitting your manuscript to PLOS ONE. After careful consideration, we feel that it has merit but does not fully meet PLOS ONE’s publication criteria as it currently stands. Therefore, we invite you to submit a revised version of the manuscript that addresses the points raised during the review process. Can you please address a few outstanding concerns raised by the expert review?

We look forward to receiving your revised manuscript.

Kind regards,

Avanti Dey, PhD

Staff Editor

PLOS ONE

Journal Requirements:

Reviewers' comments:

Reviewer's Responses to Questions

**Comments to the Author**

1. If the authors have adequately addressed your comments raised in a previous round of review and you feel that this manuscript is now acceptable for publication, you may indicate that here to bypass the “Comments to the Author” section, enter your conflict of interest statement in the “Confidential to Editor” section, and submit your "Accept" recommendation.

Reviewer #1: All comments have been addressed

2. Is the manuscript technically sound, and do the data support the conclusions?

Reviewer #1: Yes

3. Has the statistical analysis been performed appropriately and rigorously? 

Reviewer #1: Yes

4. Have the authors made all data underlying the findings in their manuscript fully available?

Reviewer #1: Yes

5. Is the manuscript presented in an intelligible fashion and written in standard English?

Reviewer #1: Yes

6. Review Comments to the Author

Reviewer #1: I am happy to accept this paper, but I feel addressing the following short points will serve to improve it.

- The (n=2) on line 118 looks like it can be removed now.

- On line 135, does the author want to replace "Quantitively analysis" with "Quantitative analysis"?

- In the supp material, the value on line 49 of 0.05 might need to be updated, perhaps they mean 0.95?

- In the supp material line 70, if possible, increase the resolution of this image.

- In addressing feedback I gave before, a section of the paper was removed and is now in the supplementary information. Now all references to the Kappa score have been removed from the paper. It would be useful to have one or two lines in the paper to acknowledge what these scores tell about the level of agreement between reviewers, rather than having readers search the supplementary information. It will also look more convincing to some readers. Maybe this would go in the "Quantitively analysis" (which might need to be renamed as mentioned above) section before the "Results". But this is up to the author.

Thank you again for the interesting read.

7. PLOS authors have the option to publish the peer review history of their article (what does this mean?). If published, this will include your full peer review and any attached files.

Reviewer #1: No

---

## [Author Response · Author response to Decision Letter 1]

30 Aug 2022

We greatly appreciate the feedback from this reviewer! We applaud their commitment and approach to providing our reviews and feel that this manuscript has been improved thanks to their recommendations.

---

## [Decision Letter · Decision Letter 2]

8 Sep 2022

PONE-D-22-04237R2Differences in self-reported benefits for student-artist versus faculty experiences in a virtual artist-in-residence program.PLOS ONE

Dear Dr. Oliver,

Thank you for submitting your manuscript to PLOS ONE. After careful consideration, we feel that it has merit but does not fully meet PLOS ONE’s publication criteria as it currently stands. Therefore, we invite you to submit a revised version of the manuscript that addresses the points raised during the review process.

We look forward to receiving your revised manuscript.

Kind regards,

Anand Nayyar, Ph.D.

Academic Editor

PLOS ONE

Journal Requirements:

Additional Editor Comments:

The Paper needs revisions and is subject to re-review.

Reviewers' comments:

Reviewer's Responses to Questions

**Comments to the Author**

1. If the authors have adequately addressed your comments raised in a previous round of review and you feel that this manuscript is now acceptable for publication, you may indicate that here to bypass the “Comments to the Author” section, enter your conflict of interest statement in the “Confidential to Editor” section, and submit your "Accept" recommendation.

Reviewer #1: All comments have been addressed

Reviewer #2: All comments have been addressed

2. Is the manuscript technically sound, and do the data support the conclusions?

Reviewer #1: Yes

Reviewer #2: (No Response)

3. Has the statistical analysis been performed appropriately and rigorously? 

Reviewer #1: Yes

Reviewer #2: (No Response)

4. Have the authors made all data underlying the findings in their manuscript fully available?

Reviewer #1: Yes

Reviewer #2: (No Response)

5. Is the manuscript presented in an intelligible fashion and written in standard English?

Reviewer #1: Yes

Reviewer #2: (No Response)

6. Review Comments to the Author

Reviewer #1: This is a good paper and should prove valuable. I have little more to add beyone what I have previously said. Thanks again for the intersting read!

Reviewer #2: The subject of the article is incomprehensible. Whatever the title, I don't know what the manuscript will be about. It needs to be improved. Abstract for complete reconstruction. Sentence: "The value of science communication in engaging the public has been well established." It is completely unfounded. The authors did not justify the importance of the work or its purpose. Unexplained abbreviations are used. It is unknown when the research was conducted on which sample and what results were obtained. Introduction. This point needs some fine-tuning. The authors justify the validity of the research in an interesting way, but do not state whether their proposal differs from other studies, what is novel in the work, and what gap is filled by the manuscript. I also recommend that you briefly describe what the AiR program is and why this program is important in terms of the topic under consideration. In the introduction, I also recommend that you briefly discuss how the manuscript is structured. The methodology, results, and conclusion are the strengths of the work. A weak point, however, is a very poor review of the literature, which requires a thorough elaboration. Figures 2b, c, and 3a, bc in the work are completely illegible.

7. PLOS authors have the option to publish the peer review history of their article (what does this mean?). If published, this will include your full peer review and any attached files.

Reviewer #1: No

Reviewer #2: No

---

## [Author Response · Author response to Decision Letter 2]

16 Nov 2022

We have address all comments from the reviewer to the best of our ability and have now included the correction from the editorial team (including adding the table as part of your main manuscript and remove the individual files). Detail about these changes are in the rebuttal letter. 

Thank you very much for your review and consideration of this manuscript, 

Kendra

---

## [Decision Letter · Decision Letter 3]

2 Dec 2022

How to design an art-science program? Self-reported benefits for artists and scientists in the VI4 Artist-in-Residence program

PONE-D-22-04237R3

Dear Dr. Oliver,

We’re pleased to inform you that your manuscript has been judged scientifically suitable for publication and will be formally accepted for publication once it meets all outstanding technical requirements.

Kind regards,

Anand Nayyar, Ph.D.

Academic Editor

PLOS ONE

Additional Editor Comments (optional):

The Revised Paper stands Accepted with no further revisions.

Reviewers' comments:

Reviewer's Responses to Questions

**Comments to the Author**

1. If the authors have adequately addressed your comments raised in a previous round of review and you feel that this manuscript is now acceptable for publication, you may indicate that here to bypass the “Comments to the Author” section, enter your conflict of interest statement in the “Confidential to Editor” section, and submit your "Accept" recommendation.

Reviewer #1: All comments have been addressed

Reviewer #2: All comments have been addressed

2. Is the manuscript technically sound, and do the data support the conclusions?

Reviewer #1: Yes

Reviewer #2: Yes

3. Has the statistical analysis been performed appropriately and rigorously? 

Reviewer #1: Yes

Reviewer #2: Yes

4. Have the authors made all data underlying the findings in their manuscript fully available?

Reviewer #1: Yes

Reviewer #2: Yes

5. Is the manuscript presented in an intelligible fashion and written in standard English?

Reviewer #1: Yes

Reviewer #2: Yes

6. Review Comments to the Author

Reviewer #1: (No Response)

Reviewer #2: The manuscript has been adapted to the comments of the reviewers. Thank you. The drawings are legible, and all comments have been taken into account.

7. PLOS authors have the option to publish the peer review history of their article (what does this mean?). If published, this will include your full peer review and any attached files.

Reviewer #1: No

Reviewer #2: No

---

## [Editor Report · Acceptance letter]

20 Dec 2022

PONE-D-22-04237R3 

How to design an art-science program? Self-reported benefits for artists and scientists in the VI4 Artist-in-Residence program 

Dear Dr. Oliver:

I'm pleased to inform you that your manuscript has been deemed suitable for publication in PLOS ONE. Congratulations! Your manuscript is now with our production department. 

Kind regards, 

on behalf of

Dr. Anand Nayyar 

Academic Editor

PLOS ONE